# Dynamic Changes of Ascorbic Acid, Phenolics Biosynthesis and Antioxidant Activities in Mung Beans (*Vigna radiata*) until Maturation

**DOI:** 10.3390/plants8030075

**Published:** 2019-03-25

**Authors:** Yanyan Lu, Xiaoxiao Chang, Xinbo Guo

**Affiliations:** 1School of Food Science and Engineering, South China University of Technology, Guangzhou 510641, China; yanyanl333@163.com; 2Overseas Expertise Introduction Center for Discipline Innovation of Food Nutrition and Human Health (111 Center), Guangzhou 510641, China; 3Institute of Fruit Tree Research, Guangdong Academy of Agricultural Sciences, Guangzhou 510640, China; xxchang6@163.com; 4Key laboratory of South Subtropical Fruit Biology and Genetic Resource Utilization (MOA), Guangzhou 510640, China; 5Guangdong Province Key Laboratory of Tropical and Subtropical Fruit Tree Research, Guangzhou 510640, China

**Keywords:** legume development, secondary metabolites, gene expression, antioxidant activity

## Abstract

To better understand the regulatory mechanism of phenolics and ascorbic acid accumulation as well as antioxidant activities in mung beans during legume development, the gene expression profiles of 25 key-coding genes in ascorbic acid and phenolics metabolic pathways were analyzed. As well as the dynamitic changes of ascorbic acid, phenolic profiles and antioxidant activities with legume development were studied. The results indicated that gene expression profiles were closely related to the ascorbic acid and phenolics accumulation regularity during legume development. *_Vr_VTC2* and *_Vr_GME* played important roles for ascorbic acid accumulation from 8 to 17 days after flowering (DAF). *_Vr_PAL* and *_Vr_CHS* exhibited positive correlations with daidzein and glycitin accumulation, and *_Vr_IFS* had a strong positive correlation with glycitin biosynthesis. Antioxidant activities dramatically increased during mung bean maturing, which were significantly related to ascorbic acid and phenolics accumulation. Eight days after flowering was the essential stage for ascorbic acid and phenolics biosynthesis in mung beans.

## 1. Introduction

Previous studies have shown that regular consumption of fruits and vegetables lowers the risk of chronic diseases [1,2]. Mung bean has been reported as a dietary source rich in isoflavones, phenolic acids, tannis, phytic acid and vitamins [3]. Phenolics in mung beans appear to exhibit the primary function of antioxidants and are prevalent today due to their nutritional and therapeutic properties [4]. Additionally, bioactive compounds in mung beans have attracted increasing scientific interest due to their roles in preventing degenerative diseases [5].

Ascorbic acid, also known as vitamin C, is a potent water-soluble antioxidant in humans and is important for the synthesis of organic components found in extracellular matrix [6]. A wide range of studies have demonstrated the great benefits of ascorbic acid on human health, especially its roles in the metabolic regulation as redox chemistry and enzymatic cofactors [7,8,9]. Ascorbic acid also plays a key role in the cellular metabolism of plants, acting as a vital antioxidant [10]. The metabolic pathway of ascorbic acid has been illustrated by many researchers [11,12]. Gene expression profiles in the ascorbic acid metabolic pathway are closely related to changes of ascorbic acid accumulation [13]. It has been reported in the literature that GDP-l-galactose phosphorylase (VTC2) plays a key regulatory role in ascorbic acid biosynthesis [14,15]. Additionally, GDP-l-galactose phosphorylase and GDP-mannose 3,5-epimerase (GME) synergistically control ascorbic acid biosynthesis [16].

Phenolics play important roles in controlling cancer and other diseases, because they have strong activity in free radical scavenging [17]. Isoflavone is an essential part of phenolics in mung beans and sprouts, mainly including daidzein, genistein, glycitin and coumestrol [5]. Due to nutritional and commercial values of phenolics, the biosynthetic pathway of phenolics has been investigated [18], which suggested gene expression profiles affected phenolics accumulation. The up-regulation of *phenylalanine lyase* (*PAL*) and *chalcone synthase* (*CHS*) in up-stream biosynthesis pathways lead to a high accumulation of phenolics in mango peel, and similar patterns have also been reported in tomato [19]. Isoflavone synthase (IFS) is the first catalyzing enzyme in the isoflavone biosynthesis pathway, which leads to the production of isoflavones in legumes [20].

In recent years, many investigations have focused on phenolics, ascorbic acid and antioxidant activities in mung beans [3,5]. However, information about the dynamic changes of phenolics and ascorbic acid during legume development is still rare. Besides, the regulatory mechanism and the molecular level of phenolics and ascorbic acid metabolism during mung bean maturing is unknown. Therefore, the intention of this study is to evaluate the related gene expression profiles of phenolics and ascorbic acid metabolic pathways, determine the accumulation of phenolics and ascorbic acid, and investigate antioxidant activities in mung beans during legume development.

## 2. Results

### 2.1. Ascorbic Acid in Mung Beans during Legume Development

The quantification of ascorbic acid was evaluated by the HPLC assay and the results are displayed in Figure 1. It could be found that ascorbic acid content increased firstly from 16.92 ± 0.16 to 18.10 ± 0.11 mg/100 g fresh weight (FW) between 5 to 8 days after flowering (DAF), and then decreased significantly (*p* < 0.05) on 11 DAF. Then, there was a significant time-dependent increase in ascorbic acid content until it reached the highest value on 17 DAF, in which the peak value was 19.91 ± 1.70 mg/100 g FW. Although ascorbic acid content decreased on 11 and 14 DAF, it generally showed an upward trend during mung bean legume development. Similarly, it has been reported that ascorbic acid content of tomatoes increased during the fruit maturing process [21]. Differently, the level of ascorbic acid increased for the first 21 days of wheat kernel development, but then it progressively decreased until it came to values approaching detectable limits during the maturation period [22]. Programmed cell death occurred in grain during kernel development, and it was reported that programmed cell death reduction was related to the reduction of ascorbic acid [23], therefore we considered the decreased of ascorbic content on 11–14 DAF may be connected to programmed cell death reduction during mung bean maturation. What deserves to be mentioned is that ascorbic acid content of mung beans found in another study was 3.1 mg/100 g DW [3], much lower than the ascorbic acid content in our study. The reason that mung bean seeds and other legumes used in other research had a low concentration of ascorbic acid could be related to the fact that the legumes were dried before the analysis [24]. In addition, the germination led to the dramatic increase of ascorbic acid content in mung beans, and the highest ascorbic acid content of mung bean sprouts on day 8 was almost 24 times higher than the initial concentration in mung bean seeds. The same significant increase pattern was found in soya bean sprouts [25,26].

Ascorbic acid works as an essential antioxidant and enzyme cofactor in plants. Experimental theories of ascorbic acid biosynthesis have been well established [11,12]. However, little is known about the regulative mechanism of ascorbic acid accumulation in mung beans during legume development. Thus, in order to determine how the ascorbic acid metabolic pathway is regulated, related key-coding genes from ascorbic acid metabolic pathway were selected for further investigation and their expression levels in mung beans during legume development are presented in Figure 2. The four accepted biosynthesis pathways are l-galactose, l-gulose, the myoinositol and d-galacturonic acid pathway. Evidence has shown that the l-galactose pathway is the main ascorbic acid biosynthesis pathway in plants [12,27]. In first step, mannose-6-phosphate isomerase (PMI) catalyzes hexose phosphates into d-Man metabolism. Then, catalytic conversion turns d-Mannose-6-p into d-Mannose-1-p by phosphomannomutase (PMM). GDP-mannose synthesis is catalyzed by mannose-1-phosphate guanylyltransferase (GMP). Next, GDP-mannose is converted to GDP-l-galactose which is catalyzed by GDP-mannose 3,5-epimerase (GME), by a reversible double epimerization. GDP-l-Galactose is converted to l-Galactose-1-p by GDP-l-galactose phosphorylase (VTC2) which is a major control enzyme in the ascorbic acid pathway. Subsequently, l-Galactose-1-p is hydrolyzed to l-Galactose by l-galactose-1-phosphate phosphatase (VTC4). The l-Galactose-1,4-lactone is formed from released l-Galactose through the oxidation by l-galactose dehydrogenase (GalDH), which is finally oxidized by l-galactonolactone dehydrogenase (GLDH), resulting in the production of ascorbic acid. The d-galacturonic acid pathway starts from d-galacturonic acid, which is reduced to l-Galactonic acid by d-galacturonate reductase (GalUR), and then it is readily converted to ascorbic acid. On the other hand, the hydrolysis of GDP-L_Guloseleads to the production of l-gulono-1,4-lactone by l-gulonolactone oxidase (GulO) in the l-gulose pathway, and subsequently into ascorbic acid. In myoinositol pathway, d-glucose is catalyzed by myo-inositol-3-phosphate synthase (MIPS) and then the myoinositol is converted to ascorbic acid. Ascorbic acid is influenced not only by its synthesis but also by recycling. Via the ascorbate-glutathione cycle, ascorbic acid can be regenerated by dehydroascorbic acid reductase (DHAR), glutathione reductase (GR) and monodehydroascorbate reductase (MDAR). In contrast, ascorbic acid is converted to monodehydroascorbate by l-ascorbate oxidase (AO) and l-ascorbate peroxidase (APX) [11]. 

As shown in Figure 2, the greatest *_Vr_PMI, _Vr_PMM, _Vr_GMP, _Vr_GME, _Vr_GalDH, _Vr_GalUR* and *_Vr_GulO* expressions were presented in mung beans during legume development on 8 DAF, while the highest *_Vr_VTC2, _Vr_VTC4* and *_Vr_GLDH* expressions were observed on 17 DAF. As for the genes in the ascorbate-glutathione cycle, the highest expressions of *_Vr_DHAR* and *_Vr_MDAR* were shown on 8 and 17 DAF, while the *_Vr_GR* and *_Vr_APX* demonstrated the greatest expressions on 11 DAF and *_Vr_AO* indicated the supreme expression on 5 DAF. In addition, Figure 3 reveals the Pearson’s correlation coefficient among the ascorbic acid content and genes on ascorbic acid biosynthesis. *_Vr_PMI, _Vr_PMM, _Vr_GME, _Vr_VTC2, _Vr_VTC4, _Vr_GLDH, _Vr_GalUR, _Vr_GulO, _Vr_AO, _Vr_DHAR* and *_Vr_MDAR* were positively correlated with ascorbic acid content in mung beans during legume development; on the contrary, *_Vr_GMP, _Vr_GalDH, _Vr_MIPS, _Vr_GR* and *_Vr_APX* were negatively correlated with ascorbic acid content. In particular, *_Vr_MIPS* exhibited significantly negative correlations with ascorbic acid content (R^2^ = −0.95, *p* < 0.05). Interestingly, *_Vr_CHI* from the phenylpropanoid biosynthetic pathway had a strong positive correlation with ascorbic acid content (R^2^ = 0.75). Yabuta demonstrated that VTC2 had a critical role in modulating ascorbic acid concentrations in kiwifruit species under abiotic stresses [15]. In addition, Zhang found that VTC2 was a key rate-limiting step in rice ascorbic acid biosynthesis [14]. The higher contents of ascorbic acid were shown on 8 and 17 DAF with no significant differences (*p* < 0.05). It showed that *_Vr_VTC2* and *_Vr_GME* have the highest expressions on the 17 and 8 DAF, respectively. Bulley and Laing put forward that ascorbic acid content appears to mainly be the result of variation in transcription of *VTC2* and *GME* in synergy [16]. We considered that high ascorbic acid content on 8 and 17 DAF in mung beans during legume development might be due to the greatest expressions of *_Vr_VTC2* and *_Vr_GME* on 8 and 17 DAF. Other genes up-regulated on 8 and 17 DAF, including *_Vr_PMI, _Vr_PMM, _Vr_GME, _Vr_VTC2, _Vr_VTC4, _Vr_GalDH*, and *_Vr_GLDH* from the L-galactose pathway; *_Vr_*GalUR from the D-galacturonic acid pathway; *_Vr_GulO* from the L-gulose pathway, may also contribute to the rise in ascorbic acid content on 8 and 17 DAF in mung beans during legume development, but they might not be the main cause. The low expression of genes in ascorbic acid biosynthesis results in the low level of ascorbic acid content on 5, 11 and 14 DAF. What is more, the highest expression of *_Vr_APX* was found on 11 DAF, we consider *_Vr_APX* seems to make more ascorbic acid convert to monodehydroascorbate in the ascorbic acid recycling pathway (ascorbate-glutathione cycle), thus ascorbic acid content came to a sudden reduction on 11 DAF. The highest expressions of *_Vr_DHAR* and *_Vr_MDAR* on 8 and 17 DAF via the ascorbate-glutathione cycle may positively correlate with the enhancement of ascorbic acid content on 8 and 17 DAF in mung beans during legume development.

### 2.2. Phenolics in Mung Beans during Legume Development

The total phenolic contents (TPC) of mung beans for five stages (5, 8, 11, 14 and 17 DAF) are shown in Table 1. TPC demonstrated a trend which was stable first, and data ranged from 312.8 ± 37.8 to 362.1 ± 19.0 mg GAE/100 g FW from 5 to 14 DAF with no significant difference (*p* < 0.05), then dropped until the lowest TPC observed on 17 DAF (286.6 ± 2.1 mg GAE/100 g FW). Compared to the TPC that ranged from 2.05 ± 0.44 to 2.38 ± 0.34 mg/g of 20 Chinese mung bean cultivars, TPC of mung beans during legume development in our study were higher [28]. Phenolics in mung beans are considered as anti-nutrient compounds and have positive health benefits [29]. The results of TPC in our study were coincident with the finding of Butsat et al. [30], TPC of the rice husk decreased in the ranged from 1.7 to 1.1 mg GAE/g from the milk grain stage to the maturity stage during grain development. Similarly, the highest TPC were found at the initial stage of development while the lowest TPC existed at the final stage of maturation of rye grains [31]. What is more, phenolics present in kernels at the time of anthesis were likely to mediate and suppress fungal infection, the concentrations of soluble phenolic acids were the highest in the immature kernel and subsequently decrease during maturity demonstrated by the in vitro inhibition study [32].

There were three phenolics (Table 1) including two isoflavones and one phenolic acid identified in mung beans with HPLC analysis. From 5 to 17 DAF, the content of daidzein showed an increasing trend, and ranged from 18.62 ± 0.31 to 25.82 ± 3.47 mg/100 g FW while the content of glycitin showed a decreasing trend, in the change from 187.3 ± 42.1 to 105.1 ± 9.0 mg/100 g FW. Gallic acid contents showed no significant difference (*p* < 0.05) in mung beans during legume development. Those three phenolics exhibited >90% contribution to TPC, which suggested daidzein, glycitin and gallic acid were the major phenolics in mung beans during legume development. The contents of daidzein and glycitin account for 50–60% of TPC, which indicated that mung beans could be a good source of isoflavone. Coincidentally, a previous study reported that isoflavones accumulate predominantly in plants of the Leguminosae family [33]. The concentrations of daidzein, glycitin and gallic acid determined in our study were similar to values reported previously in mung beans, but other major phenolics (vitexin, isovitexin, catechin, caffeic acid and ferulic acid) reported in mung beans were not detected in our study [34].

To uncover the molecular mechanism behind the changes of phenolics, we selected related key-coding genes (*_Vr_PAL, _Vr_C4H, _Vr_4CL, _Vr_CHS, _Vr_CHR, _Vr_CHI, _Vr_IFS, _Vr_IFR* and *_Vr_DFR*) from the phenolics biosynthetic pathway for further investigation and their expression levels in mung beans during legume development are presented in Figure 4. The expressions of *_Vr_PAL*, *_Vr_C4H, _Vr_CHS* and *_Vr_IFS* displayed a downtrend while *_Vr_4CL*, *_Vr_IFR, _Vr_CHR*, *_Vr_CHI* and *_Vr_DFR* displayed an uptrend during mung bean maturation. The expression of *_Vr_PAL* increased until 8 DAF, subsequently, a significant (*p* < 0.05) decline (49%) on 17 DAF. Similarly, the expression of *_Vr_C4H* had a sharp decrease (55%) between 8 and 17 DAF. Slightly different from the two genes above, the expression of *_Vr_DFR* increased until it peaked on 11 DAF and after that, it underwent a significant (*p* < 0.05) decrement on 17 DAF. In addition, the expression of *_Vr_CHS* and *_Vr_IFS* decreased during legume development (about 78% and 60% decline, respectively). On the contrary, the expression of *_Vr_4CL* and *_Vr_IFR* increased significantly (*p* < 0.05) (about 2.8 and 36.9 folds, respectively) between 5 and 17 DAF. Particularly, the higher gene expressions were observed on 8 and 17 DAF of *_Vr_CHR*, and *_Vr_CHI* low expression existed on 5, 11 and 14 DAF. 

Numerous studies have stated that the phenolics are primarily derived from the phenylpropanoid biosynthetic pathway [18,35]. As shown in Figure 4, phenylalanine lyase (PAL) is the first enzyme which removes the amine group from the amino acid and then catalyzes the conversion of phenylalanine into cinnamate. Subsequently, cinnamic acid 4-hydroxylase (C4H), one of the cytochrome P450 monooxygenases in this pathway, adds a hydroxyl group to form p-coumarate. Then, the 4-coumarate-CoA ligase (4CL) activates the coumarate by attaching a CoA at the three-carbon side chain. Next, chalcone synthase (CHS) is the key enzyme that catalyzes the reaction of p-coumaroyl CoA to chalcone. Thereafter, the chalcone synthesized by CHS can be converted to the flavanone naringenin (5,7,4′-trihydroxyflavanone) by the enzyme chalcone isomerase (CHI). Naringenin is one of the shared substrates between flavonoid and isoflavonoid pathways. The enzyme NAD(P)H-dependent 6′-deoxychalcone synthase (CHR) and CHI work together to produce isoliquiritigenin and then liquiritigenin. Liquiritigenin is the precursor for daidzein and converted to glycitin. Next, the enzyme isoflavone synthase (IFS), the key metabolic entry point for the formation of all isoflavonoids, in conjunction with CHR forms daidzein. Isoflavone reductase-like protein (IFR) is an intermediate enzyme when daidzein is converted to glyceollins. On the other branch started from naringenin, naringenin is catalyzed to dihydroflavonols, and then reduced by dihydroflavonol reductase (DFR) to flavan-3,4-diols (leucoanthocyanins). PAL is a key enzyme of the phenylpropanoid pathway and CHS is a key metabolic control point in the biosynthesis of a large number of flavonoids and isoflavonoid metabolite [35]. As shown in Figure 3, correlations among contents of phenolics and TPC and biosynthetic genes expressions in mung beans during legume development were calculated. *_Vr_PAL* and *_Vr_CHS* presented positive correlations with daidzein, glycitin and TPC while those genes had negative correlations with gallic acid content. Especially, the glycitin content exhibited significantly associated with *_Vr_PAL* (R^2^ = 0.95, *p* < 0.05) and *_Vr_CHS* (R^2^ = 0.98, *p* < 0.01). The accumulation of phenolics is demonstrated and might could be explained by the enhancement of PAL activity [36], and similarly, the increased CHS enzymatic activities lead to an increase of phenolics compounds [37]. In addition, the enzyme IFS acted as the key metabolic entry point for the formation of all isoflavonoids [38]. According to the results, *_Vr_IFS* had a strong positive correlation between glycitin (R^2^ = 0.85) and TPC (R^2^ = 0.76), but on the contrary, *_Vr_IFS* showed strong negative correlation with gallic acid content (R^2^ = −0.81). 

### 2.3. Dynamic Changes of Antioxidant Activity in Mung Beans during Legume Development

Antioxidant activity evaluation in extraction of five stages (5, 8, 11, 14 and 17 DAF) in mung beans during legume development performed by ORAC assay. As shown in Figure 5, the ORAC assay results are more accurate because human errors and frustration associated with the tedious sample preparation could be eliminated [39]. Firstly, the ORAC value increased from 5 to 8 DAF in the change from 3.77 ± 0.44 to 4.93 ± 0.42 mmol TE/100 g FW, and then the ORAC value decreased to 3.80 ± 0.64 mmol TE/100 g FW on 14 DAF. Finally, the ORAC value came to a big growth to 6.62 ± 0.35 mmol TE/100 g FW on 17 DAF, which was an almost 2-fold increase in the ORAC value of 5 DAF. 

Antioxidant activities dramatically increased during mung bean maturing, and ORAC values were consistent with the results reported by Zia-Ul-Haq et al. [40]. Their study showed ORAC values in the range of 5.64–7.23 mmol TE/g in mung beans. Ascorbic acid has a strong antioxidant potential, and the dynamic changes of ORAC values in agreement with changes of ascorbic acid content in mung beans during legume development. Similarly, there was a high linear correlation between the ascorbic acid content and the total antioxidant activity in yuzu during maturation [41]. Interestingly, unlike the study which had shown a linear relationship between TPC and antioxidant activity [42], there were no linear correlations between TPC and ORAC values in our study. However, another report showed that ORAC values had no agreement with TPC, and higher TPC were found in raspberries than in blueberries, while lower ORAC values were detected in raspberries [43]. Moreover, Figure 3 indicated that *_Vr_CHI* had a significantly high correlation with the ORAC assay (R^2^ = 0.98, *p* < 0.01), simultaneously, *_Vr_VTC2* which is the key rate-limiting gene in the ascorbic acid biosynthetic pathway demonstrated a strong positive relationship with ORAC values (R^2^ = 0.86, *p* < 0.05). The results of our study also demonstrated that mung bean could be a potentially valuable legume crop with high antioxidant potential.

## 3. Materials and Methods

### 3.1. Sample Collection

Mung bean (*Vigna radiata*) seeds were planted in the experimental field of Guangdong Academy of Agricultural Sciences (Guangzhou, China), the conditions during planting were strictly controlled. In our study we used mung beans collected 5, 8, 11, 14 and 17 days after flowering (DAF) as the experimental sample. Mung beans were frozen in the liquid nitrogen the moment immediately after they were collected, and then stored at −80 °C until analysis. 

### 3.2. Chemical and Reagents

Folin–Ciocalteu reagent, gallic acid, daidzein, glycitin, L-ascorbic acid (ASA), 2,2’-azobis-amidinopropane (ABAP), Trolox and Fluorescein disodium salt were purchased from Sigma Aldrich (St. Louis, Mo, USA). Acetone, sodium hydroxide (NaOH), methanol, ethylenediamine tetra-acetic acid (EDTA), Sodium carbonate (Na_2_CO_3_), potassium dihydrogen phosphate (KH_2_PO_4_) and dipotassium hydrogen phosphate (K_2_HPO_4_) were purchased from Sangon Biotech Co. Ltd. (Shanghai, China). Acetonitrile and methanol (HPLC grade) were purchased from ANPEL Scientific Instrument Co. (Shanghai, China). Other reagents used were analytical grade. 

### 3.3. RNA Extraction, cDNA Synthesis and Quantitative Real-Time PCR Analysis

Total RNA was extracted using HP Plant RNA Extraction Kit (OMEGA Bio-Tek, Guangzhou, China), cDNA synthesis was carried out by FastKing RT kit with gDNase (Tiangen Biotech, Beijing, China), and quantitative real-time PCR analysis was conducted with Talent qPCR PreMix with SYBR Green (Tiangen Biotech, Beijing, China) according to manufacturer’s instructions. The *ACTIN* (*Vigna radiata*) was regarded as the reference gene and the primers used in this study are listed in Table 2. Differences in relative expression levels of each gene were analyzed by the 2^−△△Ct^ method. Results are shown as mean  ±  SE (*n*  =  3).

### 3.4. Phenolics Extraction and Determination of Total Phenolic Content (TPC)

Phenolics extraction of mung beans were performed according to the method of Guo et al. with modification [26]. Data were expressed as milligram gallic acid equivalent per gram of fresh weight of mung bean (mg GAE/100 g FW). The assay was carried out in triplicate. 

### 3.5. Determination of Phenolics and Ascorbic Acid by High-Performance Liquid Chromatography (HPLC)

Phenolics and ascorbic acid analysis of mung bean extracts were executed by HPLC referring to the former study [44] and slightly modified. Briefly, phenolics and ascorbic acid were separated using C18 column (250 × 4.6 mm, 5 μm) with a temperature of 35 °C and detected by Waters 2998 Photodiode Array Detector (Waters Co., USA) at 254 and 280 nm wavelengths, respectively. The column was eluted with a gradient of 0.1% trifluoroacetic acid in water (A) and acetonitrile (B) at the flow rate of 1.0 mL/min. The gradient elution was as follows: 0–5 min (95% A), 5–40 min (95–75% A), 40–47 min (75–62% A), 47–49 min (62–55% A), 49–51 min (55% A), 51–70 min (55–20% A), 70–75 min (20–5% A), 75–77 min (5–95% A), 77–90 min (95% A). The injection volume was 30 μL. l-ascorbic acid, gallic acid, daidzein and glycitin (St. Louis, Mo, USA) were used as standards. Measured values were expressed as milligrams per 100 g of the fresh weight of mung bean (mg/100 g FW), averages were expressed from three biological replicates. HPLC profiles for ascorbic acid and phytochemicals at 254 nm in mung beans during legume development are listed in Figure 6.

### 3.6. Determination of Antioxidant Activity

The antioxidant activities of mung bean extractions were determined using oxygen radical absorbance capacity (ORAC) assay [45]. The ORAC values were measured presented as micromol equivalent of Trolox per gram in fresh weight (μmol TE/g FW). The assay was carried out in triplicate. 

### 3.7. Statistical Analysis

Experimental results were given in triplicate, and mean values ± standard deviations were reported. Statistically analyses between groups were performed using one-way analysis of variance (ANOVA) and Ducan’s multiple comparison post-test (*p* < 0.05). Means were calculated by post hoc multiple comparison method of LSD to verify whether they were significant differences (*p* < 0.05). All statistical data were analyzed by IBM SPSS software 21.0 (SPSS Inc., Chicago, IL, USA). 

## 4. Conclusions

In summary, our study revealed the changing pattern of phenolics and ascorbic acid in mung beans during legume development, as well as elucidated the molecular mechanism of these changes by researching the related gene expression profiles of phenolics and ascorbic acid biosynthetic pathway. Results showed that the higher expression of key genes of phenolics and ascorbic acid pathways led to greater accumulation of phenolics and ascorbic acid. *_Vr_VTC2* and *_Vr_GME* in the ascorbic acid biosynthetic pathway played important roles for ascorbic acid accumulation from 8 to 17 DAF. *_Vr_PAL* and *_Vr_CHS* in the phenolics biosynthetic pathway exhibited positive correlations with daidzein and glycitin accumulation. In addition, antioxidant activity had a strong correlation with ascorbic acid. The higher contents of ascorbic acid, glycitin, gallic acid and total phenolics were found at 8 DAF. Consequently, 8 DAF was considered to be the essential stage for phenolics and ascorbic acid accumulation in mung beans during legume development.

## Figures and Tables

**Figure 1 plants-08-00075-f001:**
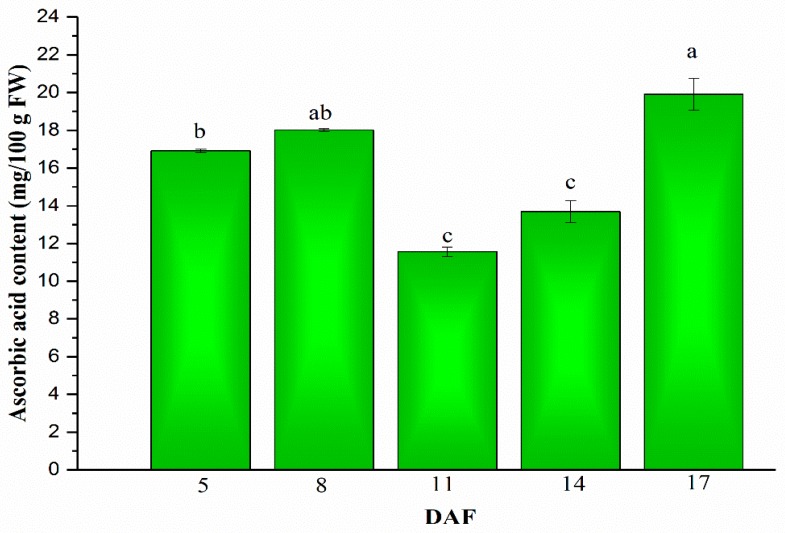
Changes of ascorbic acid content in mung beans during legume development (5, 8, 11, 14 and 17 days after flowering (DAF)). Different letters are statistically different (*p* ≤ 0.05) and represent the differences between treatments.

**Figure 2 plants-08-00075-f002:**
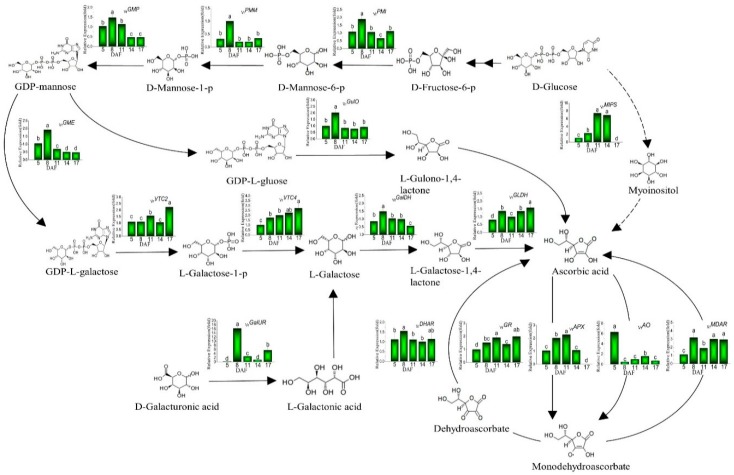
Gene expression profiles in ascorbic acid synthetic and metabolic pathways during legume development. Different letters are statistically different (*p* ≤ 0.05) and represent the differences between treatments. *PMI*: *mannose-6-phosphate isomerase*; *PMM*: *phosphomannomutase*; *GMP*: *mannose-1-phosphate guanylyltransferase*; *GME*: *GDP-mannose 3,5-epimerase*; *VTC2*: *GDP-l-galactose phosphorylase*; *VTC4*: *l-galactose-1-phosphate phosphatase*; *GalDH*: *l-galactose dehydrogenase*; *GLDH*: *l-galactonolactone dehydrogenase*; *GalUR*: *d-galacturonate reductase*; *MIPS*: *myo-inositol-3-phosphate synthase*; *GulO*: *l-gulonolactone oxidase*; *AO*: *l-ascorbate oxidase*; *APX*: *l-ascorbate peroxidase*; *GR*: *glutathione reductase*; *DHAR*: *dehydroascorbic acid reductase*; *MDAR*: *monodehydroascorbate reductase*.

**Figure 3 plants-08-00075-f003:**
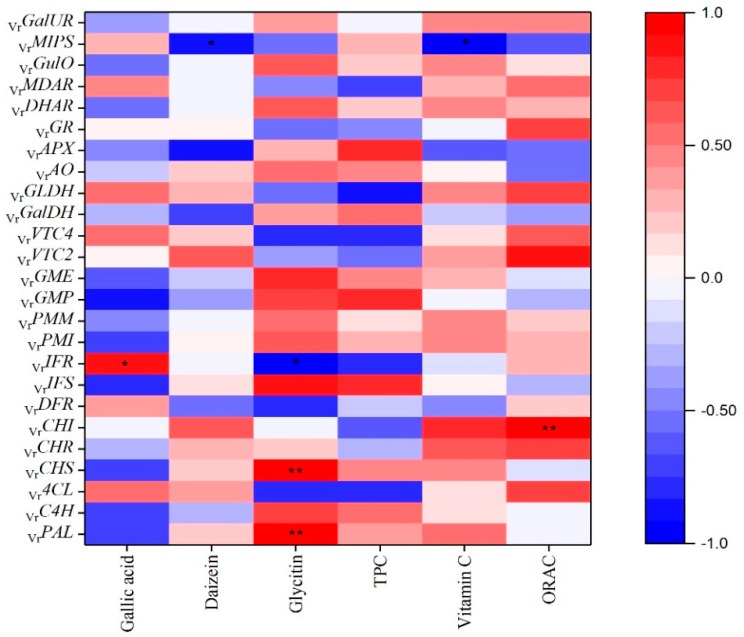
Pearson’s correlation coefficient among phytochemical profiles and antioxidant activity and genes of phenolics and ascorbic acid metabolism.

**Figure 4 plants-08-00075-f004:**
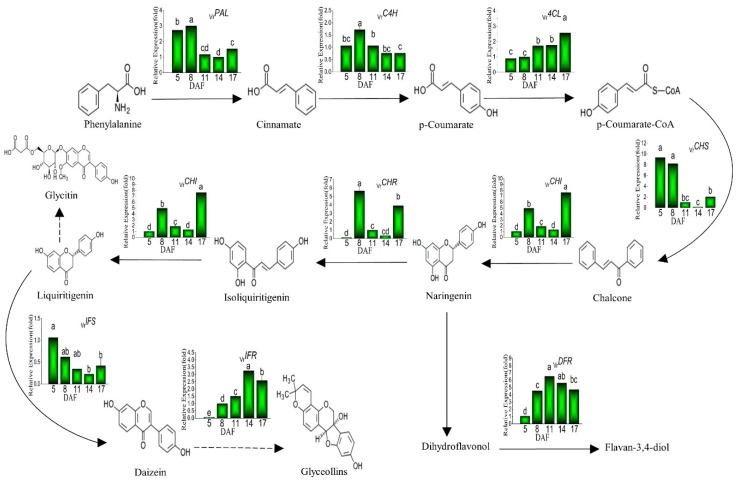
Gene expression profiles in phenolics synthetic pathway during legume development. Different letters are statistically different (*p* ≤ 0.05) and represent the differences between treatments. *PAL*: *Phenylalanine ammonia lyase*; *C4H*: *cinnamic acid 4-hydroxylase*; *4CL*: *4-coumarate-CoA ligase*; *CHS*: *chalcone synthase*; *CHR*: *NAD(P)H-dependent 6′-deoxychalcone synthase*; *CHI*: *chalcone flavanone isomerase*; *IFS*: *isoflavone synthase*; *IFR*: *isoflavone reductase-like protein*; *DFR*: *vestitone reductase*.

**Figure 5 plants-08-00075-f005:**
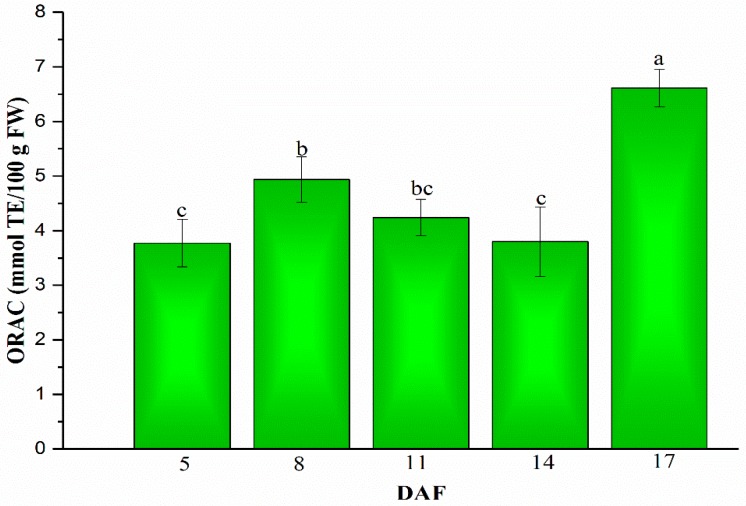
Changes of antioxidant activities in mung beans during legume development (5, 8, 11, 14 and 17 DAF). Bars with different letters differ significantly at *p* < 0.05.

**Figure 6 plants-08-00075-f006:**
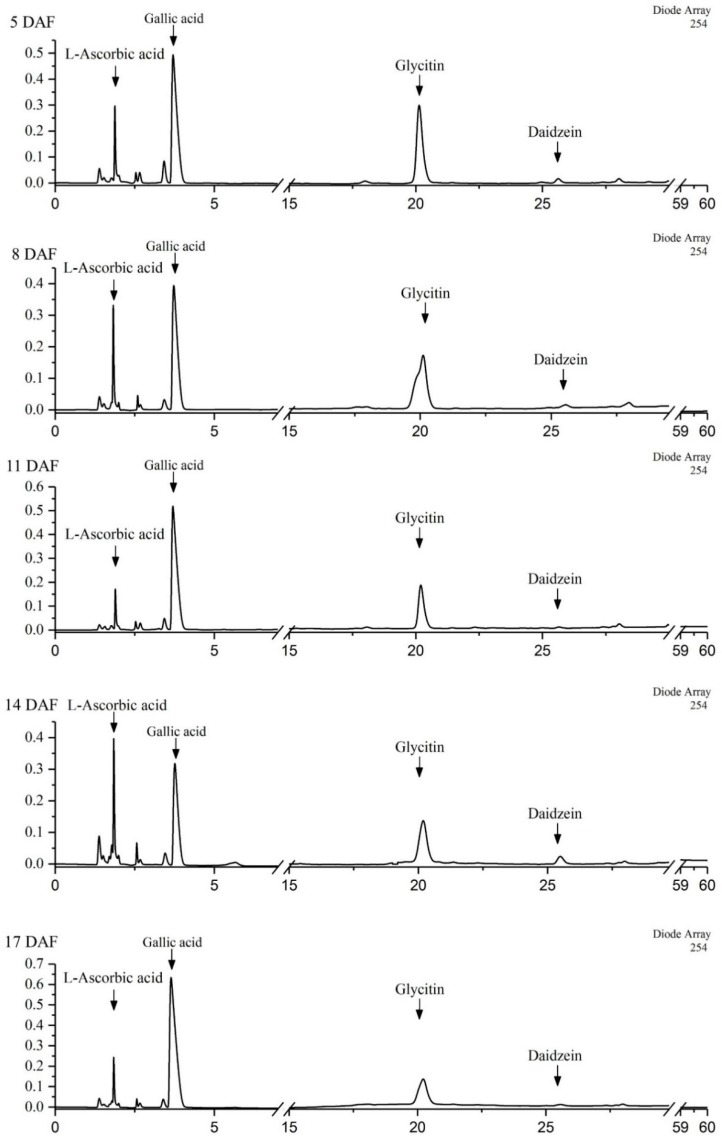
HPLC profiles for ascorbic acid and phenolics at 254 nm in mung beans during legume development.

**Table 1 plants-08-00075-t001:** The phenolics composition and contents (mg/100 g FW) in mung beans during legume development. Different letters (a–d) within the same rows indicate significant difference at *p* < 0.05.

Composition	5 DAF	8 DAF	11 DAF	14 DAF	17 DAF
Daidzein	18.62 ± 0.31 ^b^	12.12 ± 1.69 ^c^	7.73 ± 0.36 ^d^	9.45 ± 0.29 ^c,d^	25.82 ± 3.47 ^a^
Glycitin	187.3 ± 42.1 ^a^	172.3 ± 17.1 ^a^	111.9 ± 8.7 ^b^	76.53 ± 5.21 ^b^	105.1 ± 9.0 ^b^
Gallic acid	139.4 ± 52.0 ^a^	135.9 ± 46.6 ^a^	141.0 ± 52.6 ^a^	219.2 ± 34.4 ^a^	174.3 ± 89.4 ^a^
Total phenolics	362.1 ± 18.9 ^a^	346.0 ± 5.5 ^a^	364.5 ± 39.6 ^a^	312.8 ± 37.8 ^a,b^	286.6 ± 2.1 ^b^

**Table 2 plants-08-00075-t002:** Primers used in quantitative real-time PCR.

Gene Name	GeneBank ID	Prime Direction	Primer Sequence 5′ to 3′
*_Vr_PAL*	AB858431.1	Forward primer	CAACAACGGTCTGCCTTCAA
		Reward primer	TCTTGGTTGTGTTGCTCAGC
*_Vr_C4H*	NM_001317148.1	Forward primer	AGAAGACCCTCTGTTCCAGC
		Reward primer	TAGTGCTCTTCGTGCTTCCA
*_Vr_4CL*	XM_014654045.2	Forward primer	GAGGCCACAGAGAGAACCAT
		Reward primer	CTCCCGCAGCTTCATCTTTC
*_Vr_CHS*	KP164978.1	Forward primer	TGTGCTTGTCGTGTGTTCTG
		Reward primer	GCAGTCCAAACGAGCTCAAA
*_Vr_CHR*	XM_014648050.2	Forward primer	ATGGCTGCTGTCGAAATTCC
		Reward primer	CAGCAGCAGTGTCAAAGTGT
*_Vr_CHI*	NM_001317294.1	Forward primer	GCAGCCATTGAACAGTTTGC
		Reward primer	TCTCCAATACTGCAGCCGAT
*_Vr_IFS*	AF195807.1	Forward primer	ATAGCTCAGTGGCCATGGTT
		Reward primer	AAGCTCCTCGGTCAAGTCAA
*_Vr_IFR*	XM_014664599.2	Forward primer	TGTTGAATGGCAGCGAAGAG
		Reward primer	GCTCCAGAGGTCTTGAAGGT
*_Vr_DFR*	XM_014637804.1	Forward primer	GACAGAGAAGGCAGTGCTTG
		Reward primer	CTGGCCACATCATCCACATG
*_Vr_PMM*	XM_014642393.2	Forward primer	CCATTGCATTCTTGTCACGC
		Reward primer	AGGTTTCTGGGCAGCCATTA
*_Vr_GMP*	XM_014649580.2	Forward primer	GCAGTTGTGGATTCCGACTC
		Reward primer	CGTTAGTGGCCTCAACCTTG
*_Vr_GME*	XM_014663397.2	Forward primer	TACTGGCCCACTGAAAAGCT
		Reward primer	TCCACCCATATCAGCAGCAA
*_Vr_VTC2*	XM_014668416.1	Forward primer	GAAGAACGCATGCAGAGAGG
		Reward primer	CATTGTCGCTAGCCTCCAAC
*_Vr_VTC4*	XM_014638972.2	Forward primer	GTCTTCTCAGCTGGAACCCT
		Reward primer	GCCACCGTCGGAAACTTATC
*_Vr_GalDH*	XM_014650586.2	Forward primer	GCTTGTAGGCATGAAGTCCG
		Reward primer	AAGTAAACAGCGGGAGTGGA
*_Vr_GLDH*	XM_022784598.1	Forward primer	AGTCCCTTGAGTCCTGCTTC
		Reward primer	AGCCCAGTGTTCAAATGCAG
*_Vr_AO*	XM_014648870.2	Forward primer	CCACTGCCATTCTTCGCTAC
		Reward primer	TCCGTATCTCTGTTTGCCGT
*_Vr_GalUR*	XM_014647041.2	Forward primer	AGCAGTTGAACTTGGCCTTG
		Reward primer	TCCACACACCCTTCACATCA
*_Vr_DHAR*	XM_014639262.2	Forward primer	AAGGCCATTTTCTCGAGGGA
		Reward primer	TGTTCCTCGGCGGTATAGTC
*_Vr_MDAR*	XM_014654624.2	Forward primer	CACAATGGCCGCGATATCAA
		Reward primer	TGGTCACAATACAGAGGCGT
*_Vr_APX*	XM_014638724.2	Forward primer	GGCTGCTCAAACTTCCAACA
		Reward primer	CGCCGCAGTAACTACAACTC
*_Vr_GulO*	XM_014648433.2	Forward primer	TTTCGCACACCATTCCCAAG
		Reward primer	CCACCAAGCTGAACCCTCTA
*_Vr_MIPS*	EU239689.2	Forward primer	CCACTTGTTCCACCCAGTTG
		Reward primer	CCCGGAAAACATTCAAGCCA
*_Vr_GR*	XM_014658926.2	Forward primer	ATTGCCTTGGAGTTTGCTGG
		Reward primer	ACCATCAGCTGACTTCGTGA
*_Vr_PMI*	XM_014642277.2	Forward primer	GGTTCTTCGCAAATGGGGTT
		Reward primer	CACCCTTGAGCTCCTTGAGA
*_Vr_ACT*	XM_014638363.2	Forward primer	ACCACAGCTGAGCGAGAAAT
		Reward primer	ATCATGGATGGCTGGAAGAG

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
