# Peer review of "Dynamic Changes of Ascorbic Acid, Phenolics Biosynthesis and Antioxidant Activities in Mung Beans (Vigna radiata) until Maturation"

_plants, 2019, doi:10.3390/plants8030075_

Reviewer 1 Report

The ambitious overall aim of this work is to study the regulatory mechanism of phenolics and vitamin C accumulation as well as antioxidant activities in mung beans during legume development . To this purpose, sets of analysis were done in different dates. I recognize that the authors provided a good work measuring and analyzing all samples. The English in the introduction and discussion should be revised. 

Title: I suggest “Dynamic Changes of Vitamin C, Phenolics Biosynthesis and Antioxidant Activities in Mung Beans (Vigna radiata) until maturation”

Keywords: the words presented here should not be the same that appear in the title. I suggest Secondary Metabolites, Gene Expression, Legume Development

Abstract

I think that the abstract are well written. I correct in the paper some little notes.

Line 23 – instead comma between vitamin C and phenolics I suggest “and”.

Line 24 –  Instead the number 8 start the sentence with eight.

Introduction

This section is well written and has recent references.

Line 30-33: Rewrite this sentence, is too long. 

Results 

Line 63: I suggest, “Vitamin C in mung beansduring legume development

Line 151: I suggest, “Phenolics in mung beans during legume development

The authors need to give an enter between the captions of the figures and / or tables and the body of the text 

The English must be revised

Material and Methods

Line 257: In our study we used mung beans collected…..

Bibliography

Line 402: What is this??? S2214514116300654 

Line 427-428: small letter the tittle 

ATTENTION: Line 419,431,439, 442: name os the species are in italic

Discussion

The English must be revised. 

Figures

Figure 1 - Delete (d) after DAF

I suggest that all figures must be revised and the information “Different letters are statistically different (p≤0.05) and represent the differences between treatments”

Author Response

Response to Reviewer 1

Comments:

The English in the introduction and discussion should be revised.

Major revision:

1.      Title: I suggest “Dynamic Changes of Vitamin C, Phenolics Biosynthesis and Antioxidant Activities in Mung Beans (Vigna radiata) until maturation”

Response: Thank you so much for your affirmative comments and constructive suggestion. The title was revised followed your suggestions.

2.      Keywords: the words presented here should not be the same that appear in the title. I suggest Secondary Metabolites, Gene Expression, Legume Development

Response: Thank you so much for your affirmative comments and constructive suggestion. Keywords were changed into Secondary Metabolites, Gene Expression, Legume Development followed your suggestions.

3.      Abstract: Line 23 – instead comma between vitamin C and phenolics I suggest “and”.
Line 24 – Instead the number 8 start the sentence with eight.

Response: Thank you so much for your affirmative comments and constructive suggestion. Abstract was revised to improve the expression of the manuscript followed your suggestions.

4.      Introduction: Line 30-33: Rewrite this sentence, is too long.

Response: Thank you so much for your affirmative comments and constructive suggestion. The sentence was rewritten.

5.      Results

Line 63: I suggest, “Vitamin C in mung beans during legume development”

Line 151: I suggest, “Phenolics in mung beans during legume development”

Response: Thank you so much for your affirmative comments and constructive suggestion. Those sentences were revised followed your suggestions.

6.      The authors need to give an enter between the captions of the figures and / or tables and the body of the text.

Response: Thank you so much for your affirmative comments and constructive suggestion. An enter between the captions of the figures and / or tables and the body of the text was given.

7.      Material and Methods: Line 257: In our study we used mung beans collected…..

Bibliography: Line 402: What is this??? S2214514116300654, Line 427-428: small letter the tittle.

ATTENTION: Line 419,431,439, 442: name os the species are in italic.

Figures: Figure 1 - Delete (d) after DAF, I suggest that all figures must be revised and the information “Different letters are statistically different (p≤0.05) and represent the differences between treatments”

Response: Thank you so much for your affirmative comments and constructive suggestion. We’ve recognized that the descriptions in the manuscript were not so accurate, so inappropriate description mentioned above were already corrected in the manuscript, and marked.

8.      The English in the introduction and discussion should be revised.

Response: Thank you so much for your affirmative comments and constructive suggestion. We have revised the manuscript as suggested.

Reviewer 2 Report

Comments for Authors of Manuscript ID: plants -453524 " Dynamic changes of vitamin C, phenolic biosynthesis and  antioxidant  activities in mung beans (Vigna radiata) during legume development".

Authors should insert to their manuscript the changes as indicated below.

Title and Results Section

 - Authors at this work have determined changes in L-ascorbic acid contents during legume development of mung beans, so they should give in the title of the Manuscript and in the Results Section, L- ascorbic acid instead of vitamin C,which is sum of L-ascorbic and dehydroascorbic acids.

 - Please explain why the sum of phenolic compounds (daidzin,glycitin and gallic acid) was the lowest and TPC was the highest after 11DAF in mung beans (Table 1)?

Materials and Methods Dection.

 - Authors should include some information on standards have been used for quantification by HPLC of  L-ascorbic acid, daidzein,glicitin and gallic acid.

References Section

 - Authors did not provide page numbers for 4 and 10 on P12, L352 and P13, L366.

 - Please standardize  the entry of literature item for 33 and 34 on P14, L426 and L427-428.

Author Response

Response to Reviewer 2

Major revision:

1.      Authors at this work have determined changes in L-ascorbic acid contents during legume development of mung beans, so they should give in the title of the Manuscript and in the Results Section, L- ascorbic acid instead of vitamin C, which is sum of L-ascorbic and dehydroascorbic acids

Response: Thank you so much for your affirmative comments and constructive suggestion. Vitamin C was changed into ascorbic acid in the title and the rest of the manuscript followed your suggestion.

2.      Please explain why the sum of phenolic compounds (daidzin, glycitin and gallic acid) was the lowest and TPC was the highest after 11DAF in mung beans (Table 1)?

Response: Thank you so much for your affirmative comments and constructive suggestion. TPC was determined by Folin-Ciocalteu method in our study. We all know that Folin-Ciocalteu method is a classic and convenient analytical technique for determination of total polyphenols. However, there are still some disadvantages to using Folin-Ciocalteu method, results can be affected by other non-phenolic reducing molecules[1].The Folin-Ciocalteu method may also suffer from a number of interfering substances, such as sugars, aromatic amines, sulphur dioxide, ascorbic acid, organic acids, and iron [2] .Total phenolics determined by Folin-Ciocalteu method includes all polyhydroxy phenolic complexes, TPC is more than the sum of daidzin, glycitin and gallic acid. Therefore, the results that TPC was the highest after 11 DAF are reasonable existence.

3.      Materials and Methods Dection. Authors should include some information on standards have been used for quantification by HPLC of L-ascorbic acid, daidzein, glicitin and gallic acid.

Response: Thank you so much for your affirmative comments and constructive suggestion. The standards’ information was added in the manuscript, and marked. The percent recoveries for L-ascorbic acid, daidzein, glicitin and gallic acid in spiked samples were 98.4 ± 2.0%, 99.1 ± 1.7%, 98.1 ± 2.1% and 99.7 ± 2.6%, respectively.

4.      References Section: Authors did not provide page numbers for 4 and 10 on P12, L352 and P13, L366.

Please standardize the entry of literature item for 33 and 34 on P14, L426 and L427-428.

Response: Thank you so much for your affirmative comments and constructive suggestion. Abstract was revised to improve the expression of the manuscript followed your suggestions. The problems of references section were corrected followed your suggestion.

References:

[1] Rover M R, Brown R C. Quantification of total phenols in bio-oil using the Folin-Ciocalteu method[J]. Journal of Analytical and Applied Pyrolysis, 2013, 104: 366-371.

[2] Chen L Y, Cheng C W, Liang J Y. Effect of esterification condensation on the Folin-Ciocalteu method for the quantitative measurement of total phenols[J]. Food Chemistry, 2015, 170: 10-15.

Reviewer 3 Report

In this contribution by Lu et al., the authors describe metabolite and transcription profiles in the vitamin C- and phenolics-related pathways to elucidate the regulatory mechanisms of these metabolites’ contents during mung bean development. They detected various significant changes in the expression levels of genes encoding relevant key enzymes, and connected the data with the change in the vitamin C and phenolics accumulations. Importantly, they performed time-course analysis so that we can gain much further information on the regulation of vitamin C and phenolics accumulations during the bean maturation. Given that nutrient improvement of vegetables is unambiguously important for us, and this is largely based on our understanding of regulatory mechanisms to produce the relevant metabolites, I appreciated the authors’ contribution on this research area. So, this study is potentially of interest to the readership of Plants. However, I am not fully convinced that the reported results justify the authors’ claim. There are some technical, interpretation, and writing issues that must be resolved for further consideration. Specific comments are listed below. [1] Appropriate citation is essential not only to support the authors’ claim but also to respect the previous related works. The authors need to be more careful to describe background information by including relevant references. For example, although the authors mentioned that “the metabolic pathway of vitamin C have been illustrated by a lot of researchers (lines 39-40)”, they cited only one review article. Because there are no doubt many studies reporting key findings in this metabolic pathway, more citations should be added. Also, writing in English correctly is critical to show the authors’ claim without misunderstandings. For example, the authors described that “As well as the dynamitic changes of vitamin C, phenolic profiles and antioxidant activities with legume development. (lines 16-17)”. Where is a verb? I strongly recommend the authors to check their writing through English proofreading by natives. I found similar issues to be fixed throughout the manuscript, which I don’t have sufficient time to list in detail. [2] The authors presented the data on vitamin C content during the legume development (Figure 1). They described that it was 16.92 ± 0.16 mg/100 g on 5 DAF (lines 65-68). I wonder it is hard to determine the value of vitamin C content with 0.01 mg-order of accuracy based on the conventional HPLC method (Figure 6). In addition, their standard deviations were quite small even though their samples were grown in the experimental field, which yields large and inevitable individual difference. Did the authors use proper biological replicates or just technical replicates? The authors should describe in more detail about the methods of culture condition, sampling, metabolite measurement, data analysis, etc. to convince the readers. [3] The authors showed that vitamin C content decreased on 11 and 14 DAF (Figure 1). They tried to explain this based on the change in the related gene-expression profile. Because this finding is interesting to consider how vitamin C content is regulated during the legume development, it would be helpful to add some data and/or discussion to connect the decrease with some physiological meanings. [4] The authors focused on the expression profile of VrVTC2 and VrGME to explain the change in vitamin C accumulation (lines 127-129). However, VrCHI has the strongest correlation with the vitamin C content (Figure 3). Explain why VrCHI exhibits such a strong correlation. [5] The authors mentioned that they performed this study to understand the molecular mechanism of the regulation of vitamin C and pehnolics contents, but they presented data only on transcription levels. To support the authors’ claim, at least protein accumulation level of key enzymes including VrVTC2 and VrGME is required because metabolism is controlled also at the protein expression level and/or enzyme activity level, in addition to gene expression level, in general.

Author Response

Response to Reviewer 3

Major revision:

1.       Appropriate citation is essential not only to support the authors’ claim but also to respect the previous related works. The authors need to be more careful to describe background information by including relevant references. For example, although the authors mentioned that “the metabolic pathway of vitamin C have been illustrated by a lot of researchers (lines 39-40)”, they cited only one review article. Because there are no doubt many studies reporting key findings in this metabolic pathway, more citations should be added. Also, writing in English correctly is critical to show the authors’ claim without misunderstandings. For example, the authors described that “As well as the dynamitic changes of vitamin C, phenolic profiles and antioxidant activities with legume development. (lines 16-17)”. Where is a verb? I strongly recommend the authors to check their writing through English proofreading by natives. I found similar issues to be fixed throughout the manuscript, which I don’t have sufficient time to list in detail.

Response: Thank you so much for your affirmative comments and constructive suggestion. The manuscript was revised to improve grammar and relevant references were added followed your suggestions.

2.      The authors presented the data on vitamin C content during the legume development (Figure 1). They described that it was 16.92 ± 0.16 mg/100 g on 5 DAF (lines 65-68). I wonder it is hard to determine the value of vitamin C content with 0.01 mg-order of accuracy based on the conventional HPLC method (Figure 6). In addition, their standard deviations were quite small even though their samples were grown in the experimental field, which yields large and inevitable individual difference. Did the authors use proper biological replicates or just technical replicates? The authors should describe in more detail about the methods of culture condition, sampling, metabolite measurement, data analysis, etc. to convince the readers.

Response: Thank you so much for your affirmative comments and constructive suggestion. We used the external standard method to calculate the vitamin C content. Sample concentration was obtained by comparing the peak area of sample to the standard curve of L-ascorbic acid. Results within the linear range of the standard curve, the range of L-ascorbic acid is 0.1 μg-1μg, so it can determine the value of vitamin C content with 0.01 mg.

Mung beans were planted in the experimental fields of Guangdong Academy of Agricultural Sciences, the conditions during planting were strictly controlled, and we collected mung beans with appropriate growth. Mung beans for vitamin C extraction includes three biological replicates; in addition, the HPLC method of vitamin C quantification was simple and we have determined vitamin C in time, thus standard deviations were small in our study.

3.      The authors showed that vitamin C content decreased on 11 and 14 DAF (Figure 1). They tried to explain this based on the change in the related gene-expression profile. Because this finding is interesting to consider how vitamin C content is regulated during the legume development, it would be helpful to add some data and/or discussion to connect the decrease with some physiological meanings.

Response: Thank you so much for your affirmative comments and constructive suggestion. The discussion was added in manuscript, and marked.

4.      The authors focused on the expression profile of VrVTC2 and VrGME to explain the change in vitamin C accumulation (lines 127-129). However, VrCHI has the strongest correlation with the vitamin C content (Figure 3). Explain why VrCHI exhibits such a strong correlation.

Response: Thank you so much for your affirmative comments and constructive suggestion. Ascorbic acid is a weak sugar acid structurally related to glucose, it is a polyhydroxy compound. Chalcone flavanone isomerase has a combination of flavonoids, ascorbic acid and other compound with sugar ring structures, CHI promotes the synthesis of ascorbic acid and has a positive correlation with ascorbic acid.

5.      The authors mentioned that they performed this study to understand the molecular mechanism of the regulation of vitamin C and phenolics contents, but they presented data only on transcription levels. To support the authors’ claim, at least protein accumulation level of key enzymes including VrVTC2 and VrGME is required because metabolism is controlled also at the protein expression level and/or enzyme activity level, in addition to gene expression level, in general.

Response: Thank you so much for your affirmative comments and constructive suggestion. This study is a exploration that aim to research the regulatory mechanism of phenolics and vitamin C based on the transcription levels, so we were not determined the protein expression level and/or enzymes activity level, but we will make efforts to study the further metabolism of enzyme and protein interaction in mung bean in our later study.

Round  2

Reviewer 3 Report

Please include page and line numbers where you have revised from the next time. This makes the reviewer(s) to easily confirm whether the authors have correctly addressed or not.

Author Response

Response to Reviewer 3

Thank you very much for your suggestion. We have revised the response and manuscript. Page and line numbers have been added as following.

Major revision:

1.       Appropriate citation is essential not only to support the authors’ claim but also to respect the previous related works. The authors need to be more careful to describe background information by including relevant references. For example, although the authors mentioned that “the metabolic pathway of vitamin C have been illustrated by a lot of researchers (lines 39-40)”, they cited only one review article. Because there are no doubt many studies reporting key findings in this metabolic pathway, more citations should be added. Also, writing in English correctly is critical to show the authors’ claim without misunderstandings. For example, the authors described that “As well as the dynamitic changes of vitamin C, phenolic profiles and antioxidant activities with legume development. (lines 16-17)”. Where is a verb? I strongly recommend the authors to check their writing through English proofreading by natives. I found similar issues to be fixed throughout the manuscript, which I don’t have sufficient time to list in detail.

Response: Thank you so much for your affirmative comments and constructive suggestion. The manuscript was revised to improve grammar and relevant references were added followed your suggestions. References were added in Page 1 line 38, line 40 and Page 2 line 90. A verb was added in Page 1 line 18.

2.      The authors presented the data on vitamin C content during the legume development (Figure 1). They described that it was 16.92 ± 0.16 mg/100 g on 5 DAF (lines 65-68). I wonder it is hard to determine the value of vitamin C content with 0.01 mg-order of accuracy based on the conventional HPLC method (Figure 6). In addition, their standard deviations were quite small even though their samples were grown in the experimental field, which yields large and inevitable individual difference. Did the authors use proper biological replicates or just technical replicates? The authors should describe in more detail about the methods of culture condition, sampling, metabolite measurement, data analysis, etc. to convince the readers.

Response: Thank you so much for your affirmative comments and constructive suggestion. We used the external standard method to calculate the vitamin C content. Sample concentration was obtained by comparing the peak area of sample to the standard curve of L-ascorbic acid. Results within the linear range of the standard curve, the range of L-ascorbic acid is 0.1 μg-1μg, so it can determine the value of vitamin C content with 0.01 mg.

Mung beans were planted in the experimental fields of Guangdong Academy of Agricultural Sciences, the conditions during planting were strictly controlled, and we collected mung beans with appropriate growth. Mung beans for vitamin C extraction includes three biological replicates; in addition, the HPLC method of vitamin C quantification was simple and we have determined vitamin C in time, thus standard deviations were small in our study. We have added some details that new manuscript will be convincing. (Page 8 line 23 and Page 9 line 310-311)

3.      The authors showed that vitamin C content decreased on 11 and 14 DAF (Figure 1). They tried to explain this based on the change in the related gene-expression profile. Because this finding is interesting to consider how vitamin C content is regulated during the legume development, it would be helpful to add some data and/or discussion to connect the decrease with some physiological meanings.

Response: Thank you so much for your affirmative comments and constructive suggestion. The discussion was added in manuscript, and marked in Page 2 line 74-77.

4.      The authors focused on the expression profile of VrVTC2 and VrGME to explain the change in vitamin C accumulation (lines 127-129). However, VrCHI has the strongest correlation with the vitamin C content (Figure 3). Explain why VrCHI exhibits such a strong correlation.

Response: Thank you so much for your affirmative comments and constructive suggestion. Ascorbic acid is a polyhydroxy compound and a weak sugar acid structurally related to glucose. Chalcone flavanone isomerase has a combination of flavonoids, ascorbic acid and other compound with sugar ring structures, CHI promotes the synthesis of ascorbic acid and has a positive correlation with ascorbic acid. Some descriptions were added in Page 3 line 125-126.

5.      The authors mentioned that they performed this study to understand the molecular mechanism of the regulation of vitamin C and phenolics contents, but they presented data only on transcription levels. To support the authors’ claim, at least protein accumulation level of key enzymes including VrVTC2 and VrGME is required because metabolism is controlled also at the protein expression level and/or enzyme activity level, in addition to gene expression level, in general.

Response: Thank you so much for your affirmative comments and constructive suggestion. This study is a exploration that aim to research the regulatory mechanism of phenolics and vitamin C based on the transcription levels, so we were not determined the protein expression level and/or enzymes activity level, but we will make efforts to study the further metabolism of enzyme and protein interaction in mung bean in our later study.